# Topically Applied Resiquimod versus Imiquimod as a Potential Adjuvant in Melanoma Treatment

**DOI:** 10.3390/pharmaceutics14102076

**Published:** 2022-09-29

**Authors:** Supreeda Tambunlertchai, Sean M. Geary, Aliasger K. Salem

**Affiliations:** Department of Pharmaceutical Sciences and Experimental Therapeutics, College of Pharmacy, University of Iowa, Iowa City, IA 52242, USA

**Keywords:** melanoma, topical, imiquimod, resiquimod, adjuvant

## Abstract

Melanoma is the most lethal form of skin cancer and surgery remains the preferred and most effective treatment. Nevertheless, there are cases where surgery is not a viable method and alternative treatments are therefore adopted. One such treatment that has been tested is topical 5% imiquimod (IMQ) cream, which, although showing promise as a treatment for melanoma, has been found to have undesirable off-target effects. Resiquimod (RSQ) is an immunomodulatory molecule that can activate immune responses by binding to Toll-like receptors (TLR) 7 and 8 and may be more effective than IMQ in the context of melanoma treatment. RSQ can cross the stratum corneum (SC) easily without requiring pretreatment of the skin. In a gel formulation, RSQ has been studied as a monotherapy and adjuvant for melanoma treatment in pre-clinical studies and as an adjuvant in clinical settings. Although side effects of RSQ in gel formulation were also reported, they were never severe enough for the treatment to be suspended. In this review, we discuss the potential use of RSQ as an adjuvant for melanoma treatment.

## 1. Introduction

Melanoma is the deadliest form of skin cancer [1], typically caused by mutations in protooncogenes (e.g., B-raf) in melanocytes, the pigment-producing cells in the skin, as illustrated in Figure 1. Mutations in inherited genes such as CDKN2A and CDK4 are associated with an increased risk of melanoma development [2,3]. The 5-year survival rate of melanoma patients depends largely on the stage of the cancer at the time of diagnosis (Figure 2). For stage 0 and stage I, the 5-year survival rate is greater than 90% [4]; however, this is reduced to 50% for stage III (regional metastases) and less than 10% for stage IV (distant metastases) melanoma [5]. Currently, the preferred treatment for melanoma is surgery, which is extremely effective, especially for stage 0 and I melanoma patients. Nevertheless, there are cases where surgery is not feasible, including situations of multiple melanoma lesions on the face and neck, or patients refusing surgery. There are also cases where multiple surgical attempts fail to eradicate the disease [6,7]. Alternative treatments, such as chemotherapy, immunotherapy, and targeted therapy, are available for these cases. Among them, immunotherapy is of particular interest when it comes to improving patient survival [8,9]. It is also worth mentioning that certain immunotherapeutic molecules (e.g., CpG oligodeoxynucleotides (CpG-ODN) [10], and imiquimod (IMQ) [11,12,13]) have already been tested as topical treatments for melanoma therapy and indicate that using immunotherapeutic molecules can potentially be of benefit to melanoma patients when applied topically.

The major aim of cancer immunotherapy in general is to increase the host’s own tumor-specific immune response. Examples of agents used for this approach include IFN-α2b [14,15], antibodies to immune checkpoint proteins (e.g., anti-CTLA-4 and anti-PD-1) [15,16], IMQ [11,12,13], and resiquimod (RSQ) [17]. IMQ and RSQ represent promising topical melanoma treatment strategies, partly due to their physicochemical properties, which suit topical formulation development. Such properties include their molecular weights (Mw), being less than 500 Da [18], and their log partition coefficients (octanol-water), being in the range of 1–3 [19]. For more information, see Table 1. Although immune checkpoint inhibitors such as anti-PD1 are potent in terms of melanoma treatment efficacy [20,21], they are unsuitable for topical treatment due to their large molecular size (>140,000 Da).

**Table 1 pharmaceutics-14-02076-t001:** Physicochemical, anti-cancer related properties of IMQ and RSQ and their use in melanoma treatment.

	Imiquimod (IMQ)	Resiquimod (RSQ)
*Physicochemical properties*		
Molecular weight (Da)	240.30	314.38
Melting point (°C)	292–294	190–193
Water solubility (mg/mL)	0.247	0.312
LogP (Octanol-water)	2.83	1.55
*Anti-cancer related properties*		
Target receptor	Toll-like receptor (TLR)-7	TLR-7 and TLR-8
Pro-inflammatory activity	Yes	Yes (greater than IMQ by 10-fold)
Tumor-selective direct pro-apoptotic activity in vivo and in vitro	Yes [22]	No [22]
Effect on plasmacytoid dendritic cells (pDC)	Promotes pDC cytotoxic function against tumor cells [23]	Promotes pDC survival and viability [24]
Antiangiogenic ability	Yes [23]	No available data
Available topical (skin) formulation	Cream (Aldara™)	Gel [25]
*Uses in melanoma treatment*		
Advantages	Has direct anti-cancer properties [22,23] Evidence of successful case reports (see Table 2) Established data on the mechanisms of treatments	More potent immune response activation [24,26,27] Better skin penetration than IMQ [28] Less severe side effects (i.e., safer)
Disadvantages	Severe side effects reported [11,29] Limited skin penetration [30] Case report of de novo malignant melanoma arising after IMQ treatment [31]	Less known data on the therapeutic mechanism of treatment

**Table 2 pharmaceutics-14-02076-t002:** Case reports on using IMQ cream to treat melanoma at different stages.

Case	Melanoma Type	Dosing Regimen	Results and Tumor-Free Status	Note	Ref.
1	In situ lentigo malignant melanoma	Topically applied 5% IMQ cream to the tumor daily 5 times per week for the first 4 weeks Reduced to 2 or 3 times per week afterwards.	Completely cleared of tumor after 16 weeks of treatment. Remained tumor-free for at least 9 months.	Reduced treatment because patient experienced burning sensations and severe erosive dermatitis	[11]
2	Malignant melanoma (stage III, regional metastasis)	Topically applied 5% IMQ cream 3 times per week Applied the cream to the nodules and papules of metastatic melanoma in the evening and washed it off in the morning	Completely cleared of tumor after 4 weeks of treatment. Remained tumor-free for at least 15 months	Previously treated with wide excision and CO_2_ laser ablation	[12]
3	Malignant melanoma (stage III, regional metastasis)	Topically applied 5% IMQ cream 3 times per week Applied the cream to the nodules and papules of metastatic melanoma in the evening and washed it off in the morning	Completely cleared of tumor after 8 weeks of treatment. No report on tumor-free status	Previously treated with wide excision and CO_2_ laser ablation Found signs of erosion after 1 month of treatment but quickly healed after temporarily pausing the treatment	[12]
4	Superficially spreading malignant melanoma	Topically applied 37.5 mg 5% IMQ cream under occlusive conditions for 6–8 h twice daily for 18 weeks	Only one lesion remained after 18 weeks of treatments (the exact number of lesions prior to IMQ cream treatment was not specified) and later removed by surgery No report on tumor-free status	Previously treated with immunochemotherapy and IFN-α Refused systemic anticancer chemotherapy Erosion and side effects appeared during the treatment	[6]
5	Superficially spreading malignant melanoma	Topically applied 100 mg IMQ cream under occlusion conditions daily Reduced to once daily, 5 times per week afterwards	One lesion almost showed no response after 28 weeks of treatment (the starting number of lesions were more than one, but the exact number was not specified) No report on tumor-free status	Previously treated with immunochemotherapy and retrograde venous perfusion of liposomal doxorubicin Reduced the dosing regimen due to strong inflammation (time not stated) Experienced erosion and strong erythema over the whole treatment period (28 weeks)	[6]
6	Nodular malignant melanoma	Topically applied 25 mg IMQ cream single dose, twice daily for 6–8 h, under occlusion Added intra-lesion IL-2 therapy to the treatment after two weeks Continued with IMQ monotherapy again after the amelanotic metastases became smaller	Completely cleared of tumor after 9 weeks of treatment. No report on tumor-free status	Previously treated with multiple surgeries Refused chemotherapy Added IL-2 to the treatment because some metastatic tumors continued to grow	[6]
7	Malignant melanoma in situ	Topically applied 5% IMQ cream daily for 4 weeks Reduced the treatment to 3 times per week for 8 weeks	Completely cleared of tumor after 12 weeks of treatment. No report on tumor-free status	Refused surgical excision Developed crusting and marked erythema, therefore the treatment was paused for 1 week	[29]
8	Lentigo maligna melanoma	Topically applied 5% IMQ cream 3 times per week Applied the cream overnight on the whole cheek (site of melanoma) Continued the treatment for another 8 weeks after the patient was completely cleared of melanoma lesions	Completely cleared of tumor after 12 weeks of treatment. Remained tumor-free for at least 18 months	The lesions were too wide for excision Treatment well-tolerated	[13]
9	Lentigo maligna melanoma	Topically applied 5% IMQ cream 3 times per week Applied the cream overnight on the whole cheek (site of melanoma)	Completely cleared of tumor after 12 weeks of treatment. Remained tumor free for at least 9 months	Previously treated with at least 5 surgical excisions (with 2 previous grafts) The lesions were too wide for excision Experienced local intense inflammatory response	[13]

IMQ is a small molecule (Mw = 240.30 Da) capable of activating anti-cancer immune responses [24,32,33]. As a cream formulation, it is FDA-approved for non-melanoma treatment and has been shown to successfully treat melanoma patients at different stages [6,11,12,34]. Although these are case reports rather than systematic clinical trials capable of more powerful statistical assessments, the potential for use of IMQ or similar immunotherapeutic molecules for melanoma topical treatment is indicated. RSQ is also a small molecule (Mw = 314.40 Da) that is structurally very similar to IMQ and with a similar anti-cancer mechanism in terms of activating the immune response. However, RSQ has demonstrated greater antitumor efficacy, at least with melanoma-implanted mice [26]. Here we discuss the successful case reports on using 5% *w*/*v* IMQ cream to treat melanoma patients and the potential of using topical RSQ for the same disease.

## 2. Skin Immunity

The anti-cancer activities of both IMQ and RSQ involve the activation of immune responses via plasmacytoid dendritic cells (pDC). These immune cells are not found in normal/healthy skin but can be found in melanoma microenvironments in an inactivated state where they potentially contribute to the immunosuppressive environment within the tumor through the induction of regulatory T cells (Treg) [35,36,37]. Topical IMQ can convert these pDC to an activated state where they upregulate in the expression of type I IFN and possess increased cytotoxic function mediated by a combination of the upregulation of TRAIL (a death-inducing ligand) and the degranulation of granzyme B [23]. In addition, in a mouse model of melanoma, IMQ has been shown to recruit pDC-like cells from the blood to the skin [33]. Topical RSQ can also recruit immune cells to the skin; however, it was not clear as to which type of immune cells it can recruit [38]. The anti-cancer activity of pDC can be triggered by engaging Toll-like receptor (TLR)-7 or TLR-9 with appropriate agonists such as IMQ/RSQ or unmethylated CpG-ODN, respectively. The resultant release of type I IFN by pDC [39] promotes the activation of both innate and adaptive immune responses [23,40] and is considered essential for IMQ to mediate most of its anti-tumor properties [23,40]. More information on the tumoricidal properties of IMQ can be found in the IMQ section below.

Myeloid dendritic cells (mDC) are another important immune cell target of topical melanoma immunotherapy. There are the following 2 possible sources of mDC: those that already reside in the skin, and those that are normally in the circulatory system but can migrate to inflamed skin [41]. In humans, mDC express TLR-8 instead of TLR-7 and TLR-9, whilst in mice, TLR-8 is not functional [42]. IMQ is not an agonist for TLR-8 and therefore cannot activate mDC, while RSQ, on the other hand, can act as an agonist for both TLR-7 and TLR-8. The anti-cancer mechanisms reported to be associated with TLR-8-mediated activation of mDC include promoting the activation of tumor-specific CD8^+^ T cells, skewing CD4^+^ T cell responses towards Th1 [27], and inhibiting Treg function [27].

## 3. Imiquimod (IMQ)

### 3.1. General Information and Anti-Cancer Mechanisms

IMQ is a small compound in the imidazoquinoline group, and its physicochemical properties are listed in Table 1. At present, IMQ cream is FDA-approved for the treatment of superficial basal cell carcinoma and actinic keratosis [43,44,45]. For melanoma treatment, IMQ cream was listed as a treatment for stage III (regional metastasis) melanoma patients [46]. Although surgery is considered the gold standard for the treatment of patients with early-stage melanoma such as lentigo maligna, IMQ is now considered an effective off-label treatment alternative [47]. A clinical trial showed that IMQ can effectively treat stage 0 melanoma patients [48]. Twenty-six out of twenty-eight lentigo maligna (i.e., stage 0) patients were completely tumor-free after all the lesions were treated topically with IMQ cream daily for 3 months. Over 80% of these patients also remained tumor-free after 12 months of treatment [48]. A more recent publication (from The Netherlands) has reported on the effectiveness of using 5% IMQ cream for the topical treatment of patients with lentigo maligna, demonstrating complete clearance of the lesions in 48/57 (84.2%) of patients, with a 10.5% recurrence rate [47]. In addition, there are case reports where IMQ cream can successfully treat melanoma patients at different stages when other treatments, such as surgery, have failed (for more details see IMQ cream case studies) [11,12,13]. The efficacy of IMQ cream in melanoma treatment may be due to the ability of IMQ to activate and skew immune responses towards cellular immune responses, a type that favors the elimination of tumor cells.

The ability of IMQ to stimulate immune responses has been shown both in vitro (incubation with pDC) [24,32] and in vivo experiments [33]. Topically applying IMQ onto the skin of C57BL/6 mice can do the following: (1) recruit pDC from blood to the skin [33], (2) induce the maturation of pDC by inducing upregulation of expression of costimulatory molecules on their cell surface (e.g., CD40, CD80, and CD86) [24], (3) induce an inflammatory response in the skin and promote the migration of Langerhans cells to lymph node, and (4) up-regulate the production of different cytokines in the skin such as IFN-α and IFN-ϒ [49]. The up-regulation of these two IFNs can benefit melanoma treatment as they can skew immune responses towards being cell-mediated [50,51]. IMQ may also stimulate adaptive immune responses (with memory). In one study, the lesions of melanoma-challenged mice were treated with cryotherapy and IMQ cream (Aldara^TM^) [7]. After the first round of treatment, mice were rechallenged with melanoma cells, and it was discovered that 90% of the mice treated with the cryotherapy + IMQ cream rejected the rechallenge, while only 30% of the mice treated with cryotherapy alone were capable of rejecting the rechallenge. It was further shown that the anti-cancer activity was due to an increase in the cell-mediated immune response (increase in T cell proliferation and IFN-ϒ production).

### 3.2. Case Studies of IMQ Cream in Melanoma Patients

There are case reports on the successful use of 5% *w/v* IMQ cream to treat melanoma patients at different disease stages [11,12,13]. In one case report, two malignant melanoma patients with regional metastasis (stage III), whose surgery and laser ablation had failed to cure the disease, had their metastatic lesions topically treated with 5% IMQ cream three times per week [12]. Patients 1 and 2 were completely cleared of melanoma after 4 and 8 months of treatment, respectively, and were reported to remain tumor-free for up to 15 and 8 months, respectively (up to the time of publication) [12]. In another study, a patient with in-transit cutaneous melanoma metastases had all the lesions topically treated with 5% IMQ cream five days a week. After 6 months, all the lesions had regressed to being undetectable [23] The period in which the patient remained tumor-free was not reported. Other examples of case reports are listed in Table 2.

Treating melanoma lesions with IMQ cream results in an inflammatory response, thus implicating the involvement of the immune response [6,52,53]. However, not all tumors regress [22], potentially indicating the need for a stronger immune response in certain cases. To boost the immune response, IMQ cream has been used in combination with other treatment modalities [54]. For example, in one case study, the combination of IMQ cream and laser treatment could eradicate both local and distant metastases in the lung [54]. The IMQ cream was applied to melanoma lesions twice daily, while the laser treatment was performed directly on the skin lesions on week 2 and week 4 of the treatment course. Topical IMQ was withheld on the night of laser treatment. The results showed that skin melanoma lesions were completely cleared after 6 weeks of treatment. Interestingly, the lesions in the patient’s lungs were completely cleared 7 months after the treatment’s completion. The lung lesions were not directly treated during the 7-month period; hence, it is likely that the patient’s adaptive immune system eradicated these metastatic lesions, a phenomenon referred to as the abscopal response [55,56]. This patient remained tumor-free for at least 20 months (up to the publication date). A proposed mechanism of action of the therapeutic capacity of IMQ (and RSQ) is outlined in Figure 3.

### 3.3. Drawbacks of IMQ as a Topical Treatment for Melanoma and Future Perspectives

Despite the successful case reports and promising results from clinical trials using IMQ cream alone or as adjuvant immunotherapy to treat melanoma patients [11,12,13], there has not been general acceptance of its use in the clinic. Two possible reasons may explain this: (1) the limited skin penetration ability of IMQ, which may greatly reduce its effectiveness in curing early-stage melanoma patients, especially when compared to surgery (the accepted first-line treatment with a high success rate [46]), and (2) the possible side effects of IMQ cream. IMQ has physicochemical properties that can be considered theoretically ideal for skin penetration (e.g., molecular weight less than 500 Da [18] and logP in the range of 1–3 [19]); despite this, in practice, it has limited skin penetration ability. Pharmacokinetic studies have demonstrated that less than 1% of topically applied IMQ (Aldara™) is absorbed through the skin (package insert information Aldara™, 3M Pharmaceutical, 2004). The in vitro skin penetration study of IMQ in various solvents and pharmaceutical excipients (e.g., tween 80, polyethylene glycol, and propylene) also showed that this molecule possesses limited skin penetration ability, regardless of the excipients used [30]. In addition, IMQ cream can result in side effects (e.g., burning sensation and erosive dermatitis [11,29]), severe enough in some patients to lead to treatment suspension [12]. There have also been reports of patients developing melanoma after being treated for non-melanoma lesions with topical IMQ cream [31]. Hence, the use of topical IMQ for melanoma treatment should proceed with caution. Overall, topical application with TLR agonists such as IMQ can potentially be used for the treatment of melanoma; however, further improvements appear necessary. There are at least two ways to do this: (1) using more potent and potentially safer immunomodulatory molecules such as RSQ (RSQs side effect in clinical settings are limited and have not led to treatment suspension or adjustment [48,57], and/or (2) combining IMQ/RSQ with other treatment modalities, particularly those that directly promote tumor cell death.

## 4. Resiquimod (RSQ)

RSQ is another imidazoquinoline derivative developed in the early 1980s to treat patients infected with herpes simplex virus 2. Its chemical structure and properties are similar to IMQ (Figure 4 and Table 1).

The immune-based anti-cancer mechanisms of RSQ are similar to those of IMQ; however, RSQ has the advantage of being able to trigger both TLR-7 and TLR-8 signaling. Once administered (at the tumor site) and having penetrated into the skin/tumor, RSQ can bind to TLR-7 and TLR-8 on innate immune cells such as pDC and mDC, respectively, causing these cells to release a suite of cytokines [24], as well as to upregulate the expression of cell surface costimulatory molecules [24,58], thus enhancing immune stimulatory potency. The matured DC at the tumor site, having potentially taken up tumor antigen (e.g., from dying tumor cells) prior to maturation induction, can then migrate to the draining lymph node (due to upregulation of the chemokine receptor, CCR7) where they can initiate tumor-specific effector T cell responses [59]. These effector cells can then travel back to the cancer site, providing it has an inflammatory phenotype, and induce cancer cell death through direct antigen-specific contact. In addition, it has been shown that RSQ can reduce the immunosuppressive elements within the tumor microenvironment (TME) by decreasing the number of myeloid-derived suppressor cells (MDSC) [60] and suppressing the activity of Treg [61]. In addition, RSQ can also skew immune response towards Th1-type responses [62,63] due to its ability to induce the production of cytokines (e.g., IFN-α and IL-12) from immune cells (e.g., pDC) [24,25,51,64]. The mechanisms by which RSQ enhances cellular immune response include the following: (1) up-regulating the expression of MHC class I, (2) enhancing the responses of cytotoxic T lymphocytes (CTL) [65,66,67], and (3) polarizing CD4^+^ T cell responses towards a Th1 phenotype [27]. These properties highlight the potential for RSQ to be used as an adjuvant for cancer immunotherapy. The expression of MHC class I is necessary for tumor recognition and destruction by T cells [68,69], and the ability of CD4^+^ and CD8^+^ T cells to infiltrate tumors can have a direct correlation with the level of MHC class I expression [70].

Further evidence that topical RSQ (gel) can promote tumor-specific cell-mediated immune responses comes from both clinical and preclinical studies. A phase I clinical trial was conducted to assess the effectiveness and safety of using topical RSQ in treating patients with early-stage cutaneous T cell lymphoma (CTCL). RSQ was applied to 4–5 lesions per patient 2–7 times per week (up to 500 mg/day). Results not only showed the drug to be safe but that it promoted the recruitment of CD8^+^ T cells and NK cells to the tumor site. In addition, tumor regressions and abscopal responses were reported [38]. In a preclinical murine study, it was shown that subcutaneous (s.c.) injection of chick ovalbumin (OVA), when combined with topical application(s) of RSQ (R848), could generate significant detectable levels of OVA-specific CTL in the spleen and that these cells could protect against subsequent challenges with OVA-expressing B16 melanoma cells [25]. It is of note that topical RSQ used in melanoma treatment was also conducted in clinical trials as an adjuvant to an s.c. administered vaccine (NCT00827652 and NCT00960752). More detail about these trials can be found in the clinical studies section.

### 4.1. Resiquimod (RSQ) Versus Imiquimod (IMQ)

Both IMQ and RSQ can promote tumor regression through the activation of immune responses by acting as agonists for TLR-7 and TLR-7/TLR-8, respectively. However, RSQ has been shown to be more potent both in vitro [24] and in vivo [26]. When incubated with human pDC, 0.3 µM RSQ and 3 µM IMQ induced the secretion of the same amounts of type I interferons (IFN-α and IFN-ω) [24]. This can translate into stronger anti-cancer activity as these IFNs have direct anti-proliferative effects on tumor cells and multiple stimulatory effects on effector compartments of the immune system, such as NK cells and CTL [66,67,68]. It was also shown that RSQ can activate NFκB more potently than IMQ [24]. Since the activation of NFκB can be associated with the production of proinflammatory cytokines via the MyD88 signaling pathway [71], this may explain the ability of RSQ to induce greater cytokine production from pDC. In an in vivo study, melanoma-implanted mice were treated with IMQ and RSQ as a monotherapy (25 µg/50 µL delivered intratumorally (i.t.) 12 days post tumor challenge). A delay in tumor growth was only noticeable in the group treated with RSQ [26].

The increased potency of RSQ may be related to its capacity to trigger both TLR-7 and TLR-8 signaling whilst IMQ can only trigger TLR-7 signaling. In humans, only a small population of DC, the pDC, express TLR-7, whilst TLR-8 is expressed by the more conventional or common DC populations, such as mDC and monocyte-derived DC (mo-DC). In addition, the activation of the immune cells via TLR-8 can also result in stronger immune responses. The activation of mDC via TLR-8 by RSQ can effectively promote CD8^+^ T cell tumor recognition and polarize CD4^+^ T cells towards Th1 immune responses [27]. The same study showed that the combination of IFN-ϒ and RSQ can upregulate TLR expression on mo-DC, which potentially further amplifies the effect of TLR-8 on the activation of immune responses [27]. Additional anti-cancer mechanisms may include inhibiting Treg function by binding to TLR-8 on Treg [61] and preventing tumor-induced T cell senescence by blocking the release of cyclic adenosine monophosphate from melanoma cells [72]. Hence, the ability of RSQ to activate TLR-8 can provide advantages over IMQ, which can only activate TLR-7. This is especially true in humans because TLR-8 is functional [62,73,74], and of the DC subpopulations, only the pDC (which is a minority population) expresses TLR-7 [75].

That RSQ possesses more potent anti-cancer activities than IMQ also applies to topical formulations. IMQ cream and RSQ gel are the topical formulations normally used for treating melanoma patients [11,12,13,17,57]. From the available evidence, RSQ gel appears to possess better skin penetration ability than IMQ cream. However, the data is sparse on this topic and no side-by-side comparisons have been performed. A clinical study with healthy volunteers showed that RSQ could be detected in the serum of one of the eight subjects topically treated with 0.25% *w*/*v* RSQ gel (applied for 8 h, two times a week, a total of 3 weeks) after the last dose [28]. Although this occurred in only one of the subjects, it nevertheless suggests that RSQ in gel form may have a better skin penetrating ability compared to IMQ cream, where it was independently reported that less than 1% of the 5% *w*/*v* IMQ cream can be absorbed through the skin (package insert information Aldara™, 3M Pharmaceutical, 2004). It is worth mentioning that the concentration of RSQ gel (0.03–0.06% *w*/*v*) used in melanoma clinical trials can be lower than IMQ cream (5% *w*/*v*) by 100-fold, and there is much research into trying to improve the skin penetration of IMQ [76,77,78] and none attempting to do so for RSQ (based on an extensive literature search).

### 4.2. RSQ as a Topical Formulation

RSQ can penetrate the skin (i.e., SC) relatively easily, which is important because limited skin permeability is the biggest challenge for the development of topical therapeutic formulations. Tape stripping is a process often used to impair the SC and thereby facilitate easier access of topically applied drugs to deeper layers of the skin. CpG-ODN, a TLR-9 agonist, can induce strong cellular immune responses when topically administered [10]. It is of note that the skin needed to be pretreated with the tape stripping process in order for CpG-ODN to be effective [10]. However, tape-stripping was observed to be unnecessary for topically applied RSQ. It was shown that the level of OVA-specific CTL generated in response to a subcutaneously injected OVA vaccine combined with topically applied RSQ gel (at the vaccination site), with and without a prior tape stripping process, was comparable [25]. The levels of CTL induced by RSQ (+ vaccine) were also comparable to topically applied CpG-ODN (with tape stripping; + vaccine), and the levels of CTL induced were previously shown to have a therapeutic effect [10,79].

Topically applied RSQ can induce stronger immune responses and have reduced side effects compared to parenterally administered RSQ [80]. The small molecular weight (M_w_ = 314.38 Da) of RSQ may be the reason it disseminates rapidly to other organs after parenteral administration. While this can also be true when topically applied, the viable epidermis (epidermal layer without SC) barrier function may confine RSQ to the skin [81]. In one study, researchers tried inducing adaptive immune responses in mice by injecting OVA (intradermally) and applying RSQ gel (0.06% *w*/*v*) over the injection site in one group and at a site distal to the injection site in another group [82]. They found that RSQ only enhanced Th1 immune responses (IgG2a) when topically applied at the OVA injection site [82]. A phase I clinical study with healthy volunteers also showed that topically applied RSQ resulted in an increase in local IFN-α, an efflux of Langerhans cells, and infiltration of T cells [28]. Topically applied RSQ was also reported to have fewer side effects than parenteral (i.p.) [80]. The side effects of i.p. RSQ included influenza-like symptoms and lymphopenia [80,83]. These are believed to be the result of RSQ-inducing systemic cytokine production (e.g., IFN-α, IFN-β, and TNF-α) [80]. Administering RSQ via the i.p. route can also promote endothelial adhesiveness of leukocytes by up-regulating the adhesion molecules. This can cause the leukocytes to temporarily adhere to the vascular wall and reduce the number of leukocytes that can travel to the inflammatory site, resulting in transient immune competence [83]. These side effects were not found when RSQ was topically applied, which may be due to the limited amount of topically applied RSQ gaining access to the systemic circulation (less than 1% of the applied dose of RSQ) [28].

### 4.3. RSQ in Melanoma Treatment: Pre-Clinical Studies

RSQ can inhibit melanoma growth both in vitro and in vivo [51]. Manome et al. (2018) created a murine metastatic melanoma model using B16.F10 cells and treated them with RSQ (i.p. injection, 500 µg/dose, 3-day dosing intervals). The results showed that RSQ treatment resulted in less tumor invasion of the hindlimbs. Serum analysis also demonstrated that the levels of three cytokines (i.e., IL-6, IL-12, and IFN-ϒ) were higher in the treatment group. An in vitro study by the same group also showed that RSQ can induce higher production of these cytokines from immune cells (such as bone marrow-derived macrophages) and that all three cytokines can have anti-proliferative effects on melanoma cells [51]. The in vitro study also showed that RSQ (up to 30 ng/mL) did not have any direct effect on melanoma cells. Hence, the anticancer properties of RSQ seen in vivo are likely mediated indirectly, possibly through the induction of cytokines [51].

The anti-cancer activity of RSQ can increase significantly when used in combination with other drugs in vivo [26]. Mannan, or N-formyl-methionyl-leucyl-phenylalanine, is a phagocytosis-stimulating ligand. In one study, mannan was covalently bound to tumor (melanoma) cells and used in combination with RSQ to treat (i.t.) melanoma-implanted mice [26]. This approach was based on the rationale that the binding of the phagocytosis-stimulating ligand to the tumor cell surface will increase their uptake by DCs that infiltrate the tumor and therefore promote the presentation of tumor-associated antigen (TAA) and tumor-specific antigen (TSA) to the immune system [84,85]. The in vivo results showed that both RSQ alone (i.t. injected) and RSQ + tumor-bound mannan (i.t. injected) could delay melanoma growth compared to the untreated group and the group injected with tumor-bound mannan alone. However, only the RSQ + tumor-bound mannan group could significantly prolong survival compared to the naive group [26]. For DC to stimulate an effective tumor-specific immune response, they need the following: (1) a source of relevant antigens (TAA and/or TSA), which they take up, process, and present in association with MHC class I and class II; and (2) a danger signal, such as RSQ, that stimulates the DC to mature. Without this danger signal, DC can instead be inclined to promote tolerance to the antigens being presented.

Topical RSQ, as an adjuvant, is capable of inducing strong immune responses and can have a preventive effect against melanoma [25]. In one study, researchers immunized healthy mice with OVA (s.c.) and topically applied RSQ gel (0.2% *w*/*v*) over the vaccination site [25]. They boosted the immune response again on day 7 with OVA + RSQ gel and harvested the spleen on day 12, after the prime vaccination. The results showed that the population of OVA-specific CD8^+^ T cells found in the spleen significantly increased in the group treated with OVA + RSQ gel, compared to the groups treated with OVA alone and OVA + placebo gel. They further investigated the protective effect of these induced OVA-specific CD8^+^ T cells against OVA-expressing melanoma cells by immunizing the mice using the same aforementioned process and then challenging the mice with OVA-expressing melanoma cells on day 13 after the prime vaccination. The results showed that the group treated with OVA + RSQ gel survived significantly longer than the untreated group, indicating topical RSQ as a potential melanoma treatment.

In conclusion, RSQ can have anti-melanoma activity, which can be enhanced when used in combination with other molecules/treatments [25,26,84]. This can be true when the administration route of RSQ is topical or i.t. Hence, the idea of using topical RSQ in combination with other treatment modalities could be worth exploring in clinical studies.

### 4.4. RSQ in Melanoma Treatment: Clinical Studies

Clinical studies using topical RSQ as an adjuvant for melanoma treatment have been conducted. In one study (NCT00827652), patients with stage IIB to IV melanomas were vaccinated (s.c.) with the TAA, NY-ESO-1 protein, emulsified in Montanide (a standard oil adjuvant also containing an emulsifier) after their melanomas were surgically removed [17]. The patients were divided into two groups, one with the additional topical RSQ treatment over the vaccination site and the other without any further treatment (control group). They found that NY-ESO-1-specific antibody responses and NY-ESO-1-specific CD4^+^ T cells were induced in both groups. However, NY-ESO-1-specific CD8^+^ T cell responses, albeit small, were only found in patients treated with the vaccine + topical RSQ [86]. This study indicated that topical RSQ was safe and could help promote TAA-specific cellular immune responses in humans. The weak immune response in the absence of a tumor is similar to the results from a pre-clinical study [87] where melanoma-implanted mice were treated with high-intensity focused ultrasound (HIFU) approximately 5–7 days post tumor implantation. The tumors were surgically removed after the HIFU treatment immediately in one group and surgically removed 2 days later in another. The results showed the mice survived longer in all the HIFU treatment groups compared to the untreated group. However, the CTL activity was only enhanced in the group where the tumor was removed 2 days after the HIFU treatment, and their metastasis rates were significantly lower compared to the naïve. This suggested that the presence of a damaged tumor might be crucial for stronger immune response activation, and it was suggested by the authors of the cited work that the damaged tumor site was acting as a place for DC infiltration and maturation [87,88].

In another study (NCT00960752), patients with in-transit melanoma were intradermally treated with a TAA peptide (gp100 and MAGE-3) tumor vaccine [57]. The study groups were divided into two, one with topical RSQ (over the vaccination site) and the other without RSQ. The patients were vaccinated weekly. After 8 weeks of treatment, four out of nine patients in the topical RSQ (+ vaccine) group were completely cleared of tumor lesions compared to 0 out of 10 patients in the group receiving the vaccine alone. All the patients with remaining melanoma were further treated with topical RSQ for 16 weeks (2 times per week) for half of the remaining lesions (e.g., applying the cream to 5 out of 10 remaining lesions). Tumor regression was found in 8 out of 10 patients, 3 of whom had not received RSQ before. These two clinical studies confirmed that topical RSQ used as an adjuvant can increase TAA-based vaccine efficacy. It is worth mentioning that in both clinical studies, topical RSQ was applied at the vaccination site and not any other place on the skin. This was most likely to maximize immune responses while minimizing any possible side effects of RSQ, as the viable epidermis can act as a barrier [81] and limit the dissemination of topically applied RSQ from the skin area. Thus, the high immune response as RSQ stays in the skin area and the limited systemic side effects as limited RSQ is exposed to circulating immune cells [80,82,83]. More clinical trials (both completed and ongoing) are listed in Table 3. It is noted that the results of these trials were not posted on the clinical trials database (www.clinicaltrials.gov accessed on 14 September 2022) nor found elsewhere.

### 4.5. RSQ in Melanoma Treatment: The Potential of Topical RSQ as a Treatment for Metastatic Melanoma

The abscopal effect, where treatment of one cancer lesion can result in the regression of distal lesions, is an effect found to occur in melanoma patients and is normally associated with radiation therapy [55,89,90]. The exact underlying mechanism is still unknown; however, evidence from animal studies and patients suggests the involvement of radiation-induced systemic immune responses [89,91]. At present, there is no direct evidence showing that RSQ, in any topical form, can induce an abscopal effect. Nevertheless, there was a clinical study that showed that distant metastases can regress after using topical RSQ to treat cancer lesions (i.e., CTCL) on the skin [38]. In this study, twelve patients with early-stage CTCL were treated with between 0.03 and 0.06% *w*/*v* RSQ gel (from 4 to 5 lesions per patient were treated). It is important to note that RSQ applied at these concentrations can be considered a local treatment because the lowest concentration of RSQ gel that could induce systemic cytokine production in healthy adults is 0.25% *w*/*v* [28]. The results of the CTCL study showed that topical RSQ could induce the regression of distant, untreated skin lesions in many patients. They also showed that topical RSQ can lead to the development of systemic symptoms of low-grade fever in 2 patients (out of 12) and the rise of circulating myeloid CD80 expression (4 out of 8 tested subjects). Since topical RSQ gels containing the concentrations used here should not be able to directly induce systemic cytokine production, it is likely that these symptoms are a manifestation of locally produced cytokines that leak out into the systemic circulation. This type of ‘cytokine release’ can occasionally be seen in patients treated with topical immune response modifier therapy [48].

If topical RSQ *per se* can promote the abscopal effect, it can potentially be used to treat patients with metastatic melanoma. As yet, however, there is still insufficient evidence to support or refute the possibility. Radiation can induce the abscopal effect in melanoma patients [55,89,90]. This was believed to be associated with the ability of radiation to induce tumor-specific immune responses by promoting immunogenic tumor cell death [89,92,93]. Topical RSQ also possesses the ability to promote tumor-specific immune modulation abilities, although the actual mechanisms are different in that RSQ directly affects the immunopotency of antigen-presenting cells such as DC as opposed to having direct effects on tumor cells [60,61,62,63]. Although not tested in the context of melanoma and topical RSQ, it is possible, based on our current understanding, that topical RSQ and radiation therapy may work in synergy to combat metastatic lesions. Such a possibility is supported by findings from a study in a lymphoma model where local radiation therapy was combined with systemic RSQ and led to synergistic increases in the survival of tumor-challenged mice [94]. It is also worth mentioning that the radiation-induced abscopal effect observed in melanoma patients can be significantly enhanced when used in combination with immune checkpoint blockade (e.g., anti-CTLA4 [55,89] and anti-PD1 [90]). Combining immune checkpoint blockade with topical RSQ or with both topical RSQ and local radiation therapy may also have improved therapeutic benefits for patients with metastatic melanoma.

### 4.6. RSQ in Melanoma Treatment: Overview

Melanoma is considered a highly immunogenic cancer type, primarily due to its high mutation rate relative to other cancers. As a result, many TSAs are generated, which by definition are unique to the cancer cells in an individual patient [95,96,97,98]. Consequently, immune checkpoint blockade has managed to benefit many melanoma patients, presumably due to the activation of dormant/suppressed TSA-specific CTL, through the abrogation of PD-1:PD-L1 interactions in the TME (by anti-PD-1/PD-L1) [98,99] and possibly through the suppression of Treg function (by anti-CTLA-4) in the TME or draining lymph nodes [100,101]. Nevertheless, the success rate (i.e., complete remissions) in using immunotherapies, such as immune checkpoint blockade, has been low, and a possible explanation for this is an unfavorable TME where either other immunosuppressive factors hold sway that cannot be overcome by immune checkpoint blockade (e.g., the presence of MDSC [102]), or the TME possesses a noninflammatory phenotype and thus does not attract effector T cells to the tumor site. Topical application of RSQ to melanoma lesions has the potential to prevent the generation of tumor-induced T cell senescence [72], decrease the suppressive phenotype of MDSC [60], suppress Treg function [61], as well as generate an inflammatory TME, thus promoting recruitment of effector CTL and NK [38], and is potentially capable of collaborating with immune checkpoint blockade to promote improved antitumor immune responses. Proof of principle preclinical studies discussed above highlight this potential [26,51] as well as results from clinical studies that used topical RSQ as an adjuvant in melanoma treatment (NCT00827652, NCT00960752) as discussed in the clinical studies section.

Topical RSQ, as a monotherapy, has been shown to induce anti-melanoma immune responses in preclinical studies [26]. The anticancer activity of RSQ can be further improved when used in combination with other therapeutic strategies or when used as an adjuvant. A clinical study (NCT0096752) showed that using topical RSQ as an adjuvant to a TAA-based vaccine can lead to tumor regression in stage III melanoma patients [57]. Its potential use as an adjuvant can be extended to stage IV melanoma, but not without prior lesion excision [17]. Hence, topical RSQ can have therapeutic effects, at least when used as an adjuvant, on melanoma patients. It should be noted that adding topical RSQ did not always significantly improve the immune response [86], and more studies are still needed to understand and improve the efficacy of topical RSQ.

## 5. Conclusions

RSQ is a promising topical adjuvant for melanoma treatment based on both pre-clinical and clinical studies, and in terms of effectiveness, RSQ appears to be superior to IMQ. The topical application of RSQ has the potential to promote the regression of melanoma lesions primarily through immune-mediated mechanisms as observed through preclinical and clinical assessments. There is, however, still a paucity of clinical data evaluating the therapeutic benefit of RSQ alone and in combination with other therapies and, as such, more systematic studies are needed.

## Figures and Tables

**Figure 1 pharmaceutics-14-02076-f001:**
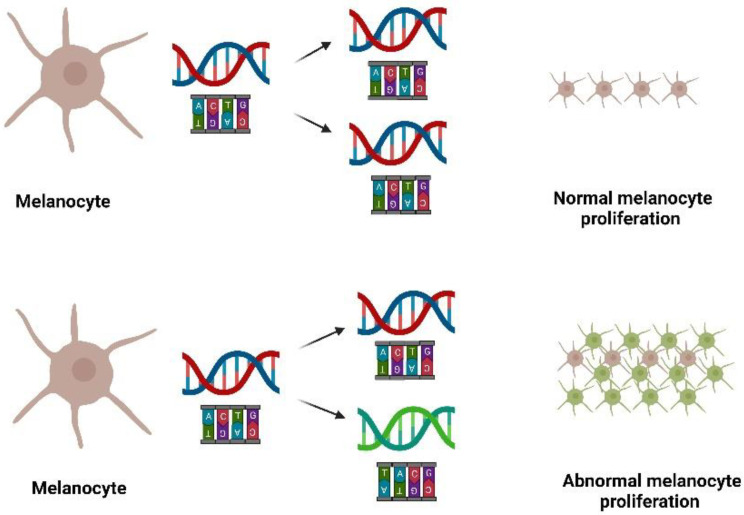
Melanoma can develop from mutations (caused by U.V. light for example; not shown) in melanocyte DNA (encoding for proto-oncogenes) that lead to dysregulated proliferation and spread of the mutated cells. Mutated DNA and melanoma cells are shown in green.

**Figure 2 pharmaceutics-14-02076-f002:**
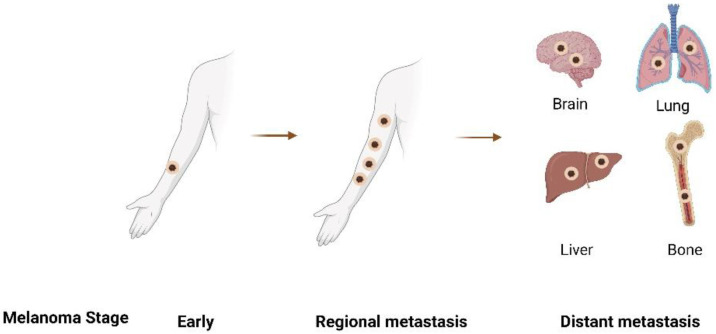
Different stages of melanoma progression.

**Figure 3 pharmaceutics-14-02076-f003:**
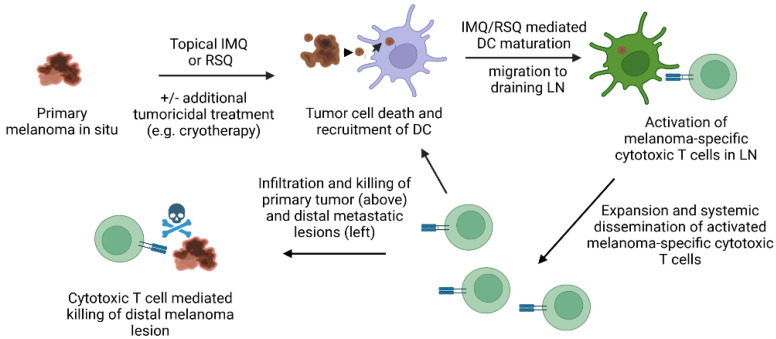
Proposed mechanism of action of IMQ/RSQ in treating melanoma. Topical treatment of in situ melanoma lesion (e.g., primary lesion) plus/minus another form of treatment (e.g., cryotherapy, ultrasound, or melanoma vaccine) leads to tumor cell death and IMQ/RSQ mediated infiltration (through TLR-mediated production of inflammatory cytokines; not shown) and activation/maturation of DC (e.g., by TLR-mediated upregulation of costimulatory molecules on the surface of DC; not shown). These DC can take up antigenic material from the dying/dead melanoma cells, travel to the draining lymph node (LN) and activate melanoma-specific cytotoxic T cell responses by presenting melanoma antigens in the context of MHC class I (not shown), likely with the aid of helper T cells (not shown). Activated melanoma-specific cytotoxic T cells then travel to cites of melanoma lesions (primary and/or metastatic lesions) that possess an inflammatory phenotype, where they can specifically recognize and kill the tumor cells. *Purple cell with brown fragment = recruited immature DC haven taken up tumor cell debris/antigen; green cell with brown fragment = activated/mature DC containing tumor cell debris/antigen; round green cells with blue receptors = melanoma-specific cytotoxic T cells + T cell receptor/TCR*.

**Figure 4 pharmaceutics-14-02076-f004:**
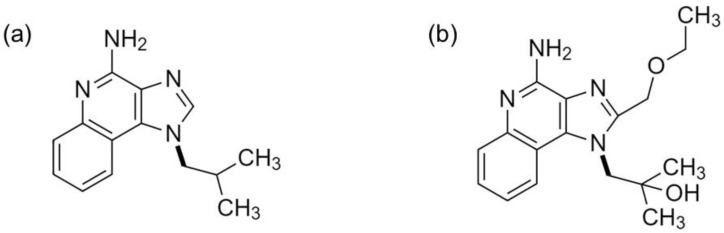
Chemical structures of (**a**) IMQ, and (**b**) RSQ.

**Table 3 pharmaceutics-14-02076-t003:** Clinical trials involving the use of topical RSQ as an adjuvant for melanoma treatment (data from www.clinicaltrials.gov accessed on 14 September 2022).

NCT Number	Title	Melanoma Condition	Vaccine/Other Adjuvants Used	Phase	Status
NCT00470379	Vaccine therapy and Resiquimod in Treating Patients with Stage II, Stage III, or Stage IV Melanoma That Has Been Completely Removed by Surgery	Completely removed stage II, III, and IV melanoma patients	NY-ESO-1b peptide vaccine	I	Completed
NCT01748747	Vaccine therapy and Resiquimod in Treating Patients with Stage II-IV, Melanoma That Has Been Completely Removed by Surgery	Completely removed stage II, III, and IV melanoma patients	Montanide ISA 51 VG, MART-1 antigen, Gag:267-274 peptide	I	Completed
NCT02126579	Phase I/II Trials of a Long Peptide Vaccine (LPV7) Plus TLR Agonists (MEL60)	Metastatic and mucosa melanoma	Peptide Vaccine (LPV7), Tetanus peptide, PolyICLC	I and II	Active
NCT00948961	A Study of CDX-1401 in Patients With Malignancies Known to Express NY-ESO-1	Advanced melanoma	CDX-1401, Poly-ICLC	I and II	Completed

## Data Availability

Not applicable.

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
