# Peer review of "Topically Applied Resiquimod versus Imiquimod as a Potential Adjuvant in Melanoma Treatment"

_pharmaceutics, 2022, doi:10.3390/pharmaceutics14102076_

Round 1

Reviewer 1 Report

The authors describe resiquimod as a promising topical adjuvant for melanoma treatment compared to imiquimod. Overall, This is an informative and useful review. I have a few minor comments, explained below.

1. Reference numbers 45 and 63, 85 and 87, 91 and 96 are the same literature, respectively. Please recheck the references.

2. Page 12, line 398: I think it's a simple mistake to apply RSQ to mice skin in clinical studies.

Reviewer 2 Report

Manuscript ID: pharmaceutics-1924317

Title: Topically applied Resiquimod as a potential adjuvant in melanoma treatment

Author: Supreeda Tambunlertchai, Sean M Geary and Aliasger K Salem

Overview and general recommendation:

The review focused on the potential use of Resiquimod as an adjuvant for melanoma treatment.

Overall, I found the review is well designed and written. However, I recommend incorporating some visualization, like the manuscript's anti-cancer mechanisms, to facilitate its reading. Also, the font size should be unified in writing the manuscript.

Reviewer 3 Report

This is a well-written review paper that summarizes the application of IMQ and RSQ in melanoma treatment via topical administration. I only have minor suggestions.

1) The title is a little bit confusing to me, shouldn't it mention IMQ if the author spent half of the paragraphs discussing that?

2) A figure to summarize the pathology, and mechanism of melanoma is appreciated.

3) A table to summarize the treatment methods, advantages and disadvantages, and application case is suggested.

4) A table to summarize RSQ for melanoma treatment cases just like for IMQ is highly suggested.
